# Are teachers missing the beat on students' motor competence?

**Fábio Flôres** [1,2,3], **Joana Serpa**[4], **Fernando Vieira**[4], **André Pombo** [5,6], **Denise Soares** [7]*, **Dimitar Shabanliyski**[7], **Rita Cordovil**[8]

**1** Universidade de Évora, Centro de Investigação em Educação e Psicologia (CIEP), Évora, Portugal, **2** Universidade de Évora, Comprehensive Health Research Centre (CHRC), Évora, Portugal, **3** Universidade de Évora, Escola de Ciências Sociais, Évora, Portugal, **4** Insight – Piaget Research Center for Ecological Human Development, Lisboa, Portugal, **5** Escola Superior de Educação, Instituto Politécnico de Lisboa, Lisboa, Portugal, **6** Research Center in Sports Performance, Recreation, Innovation and Technology (SPRINT), Lisboa, Portugal, **7** Liberal Arts Department, American University of the Middle East, Egaila, Kuwait, **8** Interdisciplinary Center for the Study of Human Performance (CIPER), Faculdade de Motricidade Humana, University of Lisbon, Lisbon, Portugal

* denise.Soares@aum.edu.kw

## Abstract

### Purpose

Compare physical education (PE) teachers' perceptions of their students' motor competence (MC) with students' objectively assessed actual motor competence.

### Methods

20 PE teachers and 340 students participated. Teachers were asked to estimate the student MC. Normative videos of the performance on each test (Standing Long Jump, Shuttle Run, Shifting Platforms, Jumping Sideways, Ball Throwing, and Kicking Velocity) were presented to assess teachers' perceptions of MC. Paired t-tests with Cohen's d quantified differences between children's actual motor competence and teachers' perceptions, alongside error tendency analyses (accurate, over-, or under-estimation) using a 5% threshold. Associations and agreement between perceived and actual MC were examined using Pearson correlations and Bland-Altman plots.

### Findings

Teachers overestimated MC in most tests, particularly stability-related tasks such as the Shifting Platforms test (p < 0.001). However, underestimations were evident in the Standing Long Jump test (p < 0.001), where students performed above national values. Also, there was a tendency to underestimate the lateral jumps, but it was not significant.

**Data availability statement:** the data is available in https://zenodo.org/ under the DOI 10.5281/zenodo.18310416.

**Funding:** The author(s) received no specific funding for this work.

**Competing interests:** The authors have declared that no competing interests exist.

## Conclusion

Our findings underscore the need for targeted teacher training programs and curriculum adjustments to improve assessment accuracy, ensuring that PE instruction effectively promotes skill development.

---

## Introduction

School strongly shapes children's motor behavior, mainly through physical education (PE). PE classes build lifelong habits of physical activity, health, and well-being, while teachers act as role models who influence students' attitudes toward fitness, movement, and sport. Teachers' understanding and assessment of each child's motor competence (MC) helps provide individualized support, boosting achievement, enjoyment, and participation beyond school. According to self-determination theory, strong motivation develops when the needs for autonomy, relatedness, and competence are met [1]. In PE classes, meeting students' need for competence seems crucial. When students feel capable in motor tasks and class activities, they show higher motivation, greater engagement, and more persistence in learning [2,3]. In Portugal, PE teachers often follow a step-by-step planning model that guides annual programs, teaching units, and daily lessons [4]. Supported by the National Physical Education Programs (PNEF) and the *Aprendizagens Essenciais*, this approach structures learning in stages across the school year to build and consolidate students' skills. Accurate diagnostic, formative, and summative assessments are essential to this planning process, as they inform each stage [5]. As a result, schools and PE teachers play a strategic role in students' behavior, underscoring the importance of effective teaching and assessment in PE.

When PE teachers acknowledge progress and offer appropriately challenging tasks, they help satisfy students' needs and strengthen their perceived competence [1]. This supports a positive self-image, increases self-determined motivation, and promotes more effective motor learning [6,7]. In contrast, negative expectations or insufficient competence-focused feedback can lower perceived competence and reduce students' engagement and effort [8]. Well-structured PE classes are crucial for fostering MC in children by offering engaging and supportive environments that promote positive experiences with movement and physical activity [9,10]. Such classes should be intentionally planned with clear objectives, as they provide diverse opportunities for children to participate in sports and motor activities vital to their development [11].

Nowadays, MC is receiving increased attention for its role in supporting lifelong physical activity. Teachers' expectations and judgments about students' future behavior or performance can powerfully shape this process [12,13]. These expectations influence teachers' attitudes, class organization, assessment practices, and student outcomes, especially in PE [12–14]. These expectations shape how teachers design learning experiences. Students perceived as high achievers often receive more instructional time, more feedback, and more encouragement, reinforcing the

teachers' original expectations through differentiated support [12]. Contemporary theoretical models emphasize that accurate feedback and appropriate task difficulty, both of which depend on valid assessment, are essential for supporting students' perceived motor competence (PMC) and self-determined motivation in PE settings [1].

As previously noted in the extant literature, MC refers to an individual's proficiency in a wide range of motor skills, namely locomotor, manipulative, and stability skills [15–17]. It is associated with mastering fundamental motor skills, which serve as a foundation for the development of specialized motor skills throughout life [18,19]. MC must also be viewed as a nonlinear progression, as each child's learning trajectory varies, with unique rates and transitions between stages [20]. MC has been investigated across various contexts and age groups [10,21,22]. Many studies have examined the association between MC and children's PMC [23–25]. However, there remains limited knowledge regarding PE teachers' PMC assessments concerning their students' actual motor competence (AMC).

MC can be assessed using different methodological approaches, which include qualitative evaluations of movement patterns and quantitative performance-based indicators (kinematic assessments). While process-oriented assessments focus on the quality and coordination of movement execution, product- or kinematic-based measures rely on objective outcomes such as time, distance, or velocity. Although these quantitative indicators are strongly associated with performance and health-related outcomes, they may lead to different interpretations of MC when considered in isolation. In PE contexts, teachers are more commonly exposed to qualitative movement observation rather than kinematic assessments; consequently, discrepancies between teachers' perceptions and objective test outcomes may occur.

Therefore, to the best of our knowledge, no previous research has examined the accuracy of PE teachers' estimations of students' MC when compared with objective, kinematic-based performance outcomes. This is a significant area of investigation, as PE teachers' perceptions may influence instructional planning and assessment decisions throughout the academic year. To address this gap, the purpose of this investigation was to compare PE teachers' perceptions of students' MC with students' AMC, as measured using standardized motor performance tests. We hypothesized that PE teachers would accurately estimate students' MC. The findings can help refine teacher training and assessment practices, ultimately enhancing the effectiveness of the PE curriculum.

## Materials and methods

### Sample

The investigation is cross-sectional and uses convenience sampling across five schools in central Portugal (during 2024 and 2025). The sample size was calculated based on the following parameters: Cohen's effect size of 0.20 for a paired-samples t-test, error probability $\alpha = 0.05$, and $\beta = 0.95$, resulting in a minimum sample size of 327 participants. The calculation was performed using GPower v3.1.9.7 [26]. Therefore, this investigation enrolled 340 participants ($13.43 \pm 2.42$ years; $19.74 \pm 4.37$ kg/m$^2$), 164 boys ($13.54 \pm 2.43$ years; $20.17 \pm 5.19$ kg/m$^2$), and 176 girls ($13.32 \pm 2.41$ years; $19.33 \pm 3.40$ kg/m$^2$). 20 PE teachers also participated in the investigation, each evaluated an average of 17 students. The selection of schools (n = 5) aimed to represent diverse educational contexts, enhancing the generalizability of findings. The inclusion criteria were assessed using information from school records and parental reports as follows: children had to be free from injuries, illnesses, or conditions that could prevent them from completing the tests, and the teachers needed to be PE teachers in the evaluated child's class.

### Procedures

The investigation occurred in two phases: (1) assessing PE teachers' perceptions of students' MC and (2) assessing children's actual MC. These assessments took place on different days to keep the evaluations independent and unaffected by one another.

The perceived MC assessment of their students (1) was conducted before administering the tests to assess AMC. All data were collected between January and March in the middle of the school year. This ensured that teachers had at least five months of contact time with their students, thereby improving their knowledge of children's motor skills.

Regarding the AMC assessment (2), data collection was conducted on regular PE class days. The assessments were conducted in a controlled setting, with no external interference (testing always occurred in a gymnasium). Before data collection, all children completed a 5-minute warm-up consisting of low- to moderate-intensity running and whole-body stretching. The test set was developed in groups of five participants per task and administered by trained examiners. A verbal explanation and a proficient demonstration of all tests were provided to all children. Children also performed a trial test before the test administration began, and all received the instruction to perform the tests to the best of their ability. No feedback regarding the test results or skill performance was provided, but motivational feedback was given to all children.

At the time of data collection, PE teachers had been working continuously with their respective classes for approximately seven to eight months (September to April). This ensured that teachers' perceptions reflected sustained pedagogical interaction and observation rather than initial or unfamiliar judgments. None of the assessments was conducted at the start of teachers' work with the students. Data were collected across two academic years, in April 2024 and April 2025. All data collection was supervised by one of the authors of this study. Parental consent and participants' oral assent were obtained before the experiment began. The University Ethics Committee approved the research (Protocol: P02-S09-27.04.22) and (Protocol: UÉ24207), and the study protocol followed the Declaration of Helsinki guidelines (2014) [27].

## Instruments

**Perceived Motor Competence assessment (PMC).** PMC assessment was evaluated through an interview with the teachers, who were asked to estimate each student's performance for each MCA test. The tests were explained and presented (through normative videos) to all teachers to evaluate their students' MC. Thus, they were asked to estimate the effect of the student's performance on the instrument's normative values [28].

Teachers' raw scores were converted to normative percentile rankings by sex and age, using six-month age intervals [28]. The stability, locomotor, and manipulative components were determined by averaging the percentile positions from the two tests for each component. Consequently, the total MC of each teacher's perception was calculated as the mean of the three MCA components. Teachers were asked to estimate students' MC using specific quantitative outcome measures following the MCA battery (e.g., time, velocity, number of repetitions). Examples of the questions are presented below (Table 1):

**Actual Motor Competence Assessment (AMC).** Children were evaluated using the Motor Competence Assessment – MCA [28,29]. The MCA instrument (Fig 1) includes six tests and three components: (1) Locomotor - Standing Long Jump (SLJ) and Shuttle Run (SHR) tests; (2) Stability - Shifting Platforms (SP) and Jumping Sideways (JS) tests; (3) Manipulative - Ball Throwing Velocity (BTV) and Ball Kicking Velocity (BKV) tests. All results were measured on a quantitative scale (i.e., distance, time, number of executions, or velocity) without a ceiling effect related to age or sex [28].

After data collection, raw scores were transformed into normative (percentile) results by sex and age in a six-month interval [28]. The stability, locomotor, and manipulative components were then calculated using the average of the respective two tests' percentile positions. Subsequently, children's total MC was calculated as the average of the three MCA components.

## Data analysis

Descriptive statistics (mean, standard deviation, and percentages) were used to describe participants. Data normality was tested with the Kolmogorov-Smirnov test, and extreme statistical outliers (identified using z-scores with ± 3.0 [30]) were checked and removed if needed [31].

Paired sample t-tests compared the children's actual MC with their PE teachers' perceptions. Effect sizes were calculated as the difference between means divided by the pooled standard deviation. The magnitude of the effect was

**Table 1. General instructions to PE teachers.**

| Category | Instructions |
| --- | --- |
| Locomotor | **SLJ general instruction**: Knowing that the student must jump, with both feet together, as far as possible, and, on average, a child of the same age and sex jumps XX centimeters, how many centimeters do you think student A will jump? |
| | **SLJ general instruction**: Knowing that the student must jump, with both feet together, as far as possible, and, on average, a child of the same age and sex jumps XX centimeters, how many centimeters do you think student A will jump? |
| Stability | **SP general instruction**: Knowing that the test takes 20 seconds and that each change of platform or the body to the platform scores 1 point, and, on average, a child of the same age and sex scores XX points, how many points do you think student A will score? |
| | **JS general instruction**: Knowing that the test takes 15 seconds and that each lateral jump scores 1 point and, on average, a child of the same age and sex scores XX points, how many points do you think student A will score? |
| Manipulative | **BTV general instruction**: Knowing that the student must throw the ball as quickly as possible and, on average, a child of the same age and sex throws at XX km/h, how many kilometers per hour will the ball reach? |
| | **BKV general instruction**: Knowing that the student must kick the ball as quickly as possible and, on average, a child of the same age and sex kicks at XX km/h, how many kilometers per hour will the ball reach? |

Note: Standing Long Jump (SLJ); Shuttle Run (SHR); Shifting Platforms (SP); Jumping Sideways (JS); (3) Ball Throwing Velocity (BTV); and Ball Kicking Velocity (BKV)

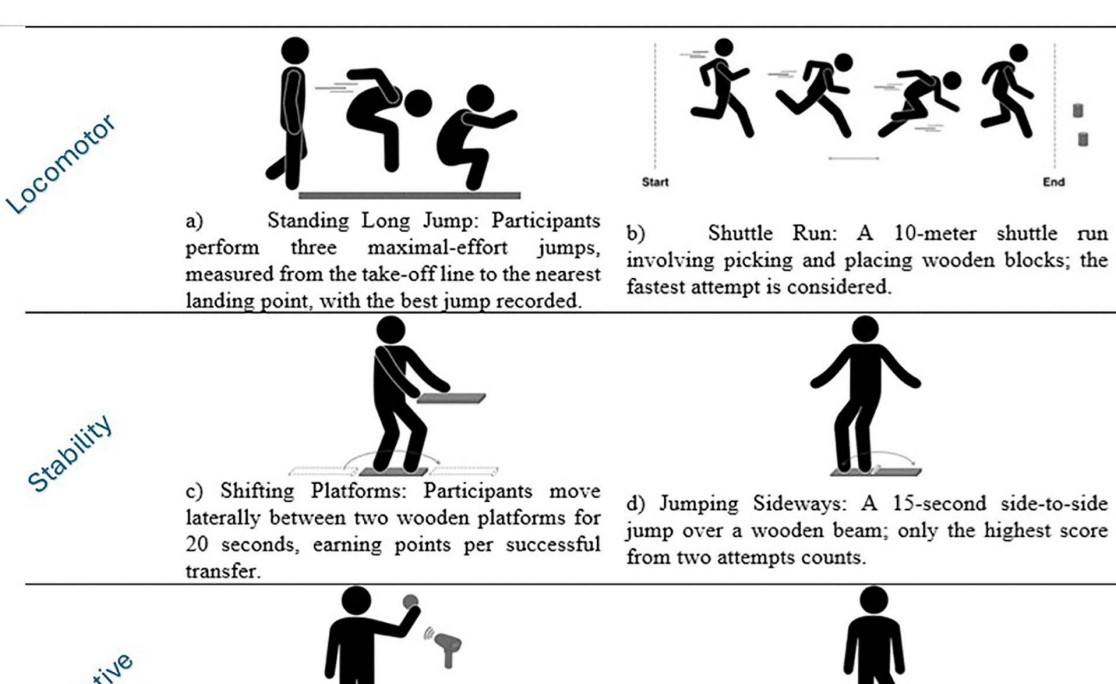

**Fig 1. Description of the Motor Competence Assessment tests and components.**

interpreted as trivial (< 0.20), small (0.20–0.49), moderate (0.50–0.79), or large (≥ 0.80), following Cohen's guidelines [32]. Error tendency (i.e., frequency of overestimation, accuracy, and underestimation bias), which indicates the direction of the error, was also calculated. The estimation was considered accurate when the difference was lower than 5%. This tolerance was adopted as a pragmatic criterion, since it is unrealistic to expect PE teachers to predict quantitative MCA outcomes (e.g., time or velocity) with zero error, and small deviations are commonly accepted in perceived research [33]. Hence, differences of more than 5% were considered overestimations if the estimate exceeded performance, or underestimations if the forecast fell short of performance. For the SHR test, the coding was inverted because when the PE teacher predicted that the student would take less time to complete the SHR than he actually did, the teacher overestimated the student's performance.

The Pearson correlation was used to assess the association between MC and RT, with coefficients < 0.30 considered weak, 0.30–0.70 moderate, and > 0.70 strong [32]. The Bland-Altman plots were generated to assess the agreement between PMC and AMC performances. The limits of agreement (mean difference ± 1.96 x SD) were also computed to interpret the extent of discrepancy between perceived and actual scores. Data analysis was performed using the Statistical Package for the Social Sciences (SPSS) version 29.0 (IBM Corp., Armonk, New York).

## Results

Table 2 summarizes the participants' sample. MC scores were differentiated in raw scores and percentiles by AMC and PMC.

Fig 2 shows notable discrepancies between AMC and PMC in almost all tests and categories. For instance, the SLJ test showed a significant underestimation of teachers' perceptions (t(339) = 5.688; p < 0.001, $\eta^2$ = 0.31). However, SP (t (339) = −17.48; p < 0.001, $\eta^2$ = − 0.95), SHR (t (339) = −8.239; p < 0.001, $\eta^2$ = 0.45), BTV (t (339) = −4.176; p < 0.001, $\eta^2$ = −0.23), BKV (t (339) = −7.205; p < 0.001, $\eta^2$ = 0.39), stability category (t (339) = −12.493; p < 0.001, $\eta^2$ = −0.68), locomotor category (t (339) = 2.020; p = 0.04, $\eta^2$ = −0.11), manipulative category (t (339) = −5.615; p < 0.001, $\eta^2$ = −0.30), and MC (t

**Table 2. Sample characterization.**

| Variables | | N | Raw Scores | | Percentiles | |
|---|---|---|---|---|---|---|
| | | | Mean | SD | Mean | SD |
| Children's general characterization | Age (years) | 340 | 13.43 | 2.42 | – | – |
| | Weight (Kg) | 340 | 49.35 | 15.12 | – | – |
| | Height (m) | 340 | 1.57 | 0.14 | – | – |
| | BMI (Kg/m$^2$) | 340 | 19.74 | 4.37 | – | – |
| AMC | JS | 340 | 33.38 | 6.64 | 37.31 | 28.39 |
| | SP | 340 | 17.80 | 6.23 | 24.63 | 30.45 |
| | SLJ | 340 | 166.52 | 34.37 | 66.59 | 27.75 |
| | SHR | 340 | 12.68 | 1.30 | 37.26 | 31.98 |
| | BTV | 340 | 13.46 | 3.34 | 41.17 | 31.81 |
| | BKV | 340 | 14.10 | 3.46 | 30.30 | 28.10 |
| PMC | JS | 340 | 30.91 | 7.78 | 35.86 | 29.33 |
| | SP | 340 | 28.19 | 9.01 | 66.65 | 30.53 |
| | SLJ | 340 | 151.25 | 30.49 | 56.35 | 26.91 |
| | SHR | 340 | 12.08 | 1.80 | 55.36 | 32.62 |
| | BTV | 340 | 13.86 | 4.44 | 50.80 | 33.10 |
| | BKV | 340 | 14.90 | 5.07 | 45.96 | 33.56 |

Note: Raw scores are measured in JS (points), SP (points), SLJ (cm), SHR (time), BTV (m/s), and BKV (m/s). All percentiles are presented in %.

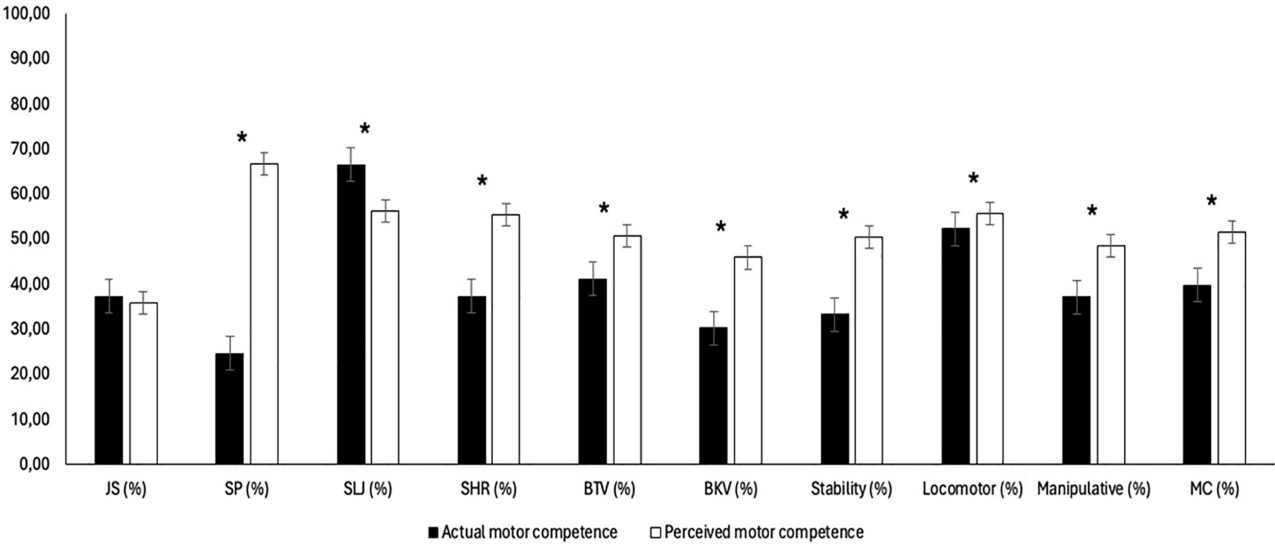

**Fig 2. Comparisons between the percentiles of AMC and PMC.** Note: *P < 0.05; Error bars represent standard deviations, illustrating the variability of scores within each assessment condition.

(339) = −8.304; p < 0.001, $\eta^2 = -0.45$) showed significant overestimations regarding PMC. No significant differences were found regarding the JS test (t (339) = 0.88; p = 0.38, $\eta^2 = 0.48$).

Table 3 presents the distribution of error tendency across the MCA tests. It was noted that there was high variability in teachers' estimates across all tests. In two tests (JS and SLJ), most teachers underestimated students' competence, whereas in the SP, SHR, BTV, and BKV, most teachers overestimated students' competence. Results showed significant differences in almost all tests (JS: $\chi^2(2) = 44.61$, p < 0.001; SP: $\chi^2(2) = 21.54$, p < 0.001; SLJ: $\chi^2(2) = 4.58$, p = 0.10; SHR: $\chi^2(2) = 10.60$, p = 0.01; BTV: $\chi^2(2) = 8.03$, p = 0.02; BKV: $\chi^2(2) = 9.31$, p = 0.01).

To further examine the agreement between teachers' PMC and students' AMC, correlation analyses and a Bland-Altman plot were conducted. As shown in Table 4, statistically significant positive correlations were observed between

**Table 3. Error Tendency by PE teachers in the different MCA tests.**

| MCA tests | | Underestimation | Accurate estimation (<5% error) | Overestimation |
|---|---|---|---|---|
| JS | % | 46.2 | 24.1 | 29.7 |
| | N | 157 | 82 | 101 |
| SP | % | 46.2 | 24.1 | 29.7 |
| | N | 157 | 82 | 101 |
| SLJ | % | 63.8 | 22.6 | 13.5 |
| | N | 217 | 77 | 46 |
| SHR | % | 53.5 | 25.3 | 21.2 |
| | N | 182 | 86 | 72 |
| BTV | % | 40.6 | 16.5 | 42.9 |
| | N | 138 | 56 | 146 |
| BKV | % | 35.6 | 17.4 | 47.1 |
| | N | 121 | 59 | 160 |

**Table 4. Associations between PMC and AMC.**

| | Stability | Stability perceived | Locomotor | Locomotor perceived | Manipulative | Manipulative perceived | AMC | PMC |
|---|---|---|---|---|---|---|---|---|
| Stability | 1 | – | – | – | – | – | – | – |
| Stability perceived | .370* | 1 | – | – | – | – | – | – |
| Locomotor | .625* | .350* | 1 | – | – | – | – | – |
| Locomotor perceived | .241* | .636* | .271* | 1 | – | – | – | – |
| Manipulative | .492* | .333* | .513* | .199* | 1 | – | – | – |
| Manipulative perceived | .037 | .412* | .022 | .497* | .165* | 1 | – | – |
| AMC | .828* | .404* | .849* | .280* | .805* | .084 | 1 | – |
| PMC | .238* | .782* | .238* | .858* | .272* | .811* | .290* | 1 |

Note: *P<0.05

AMC and PMC for all components and total scores. The correlations between actual and perceived scores were moderate for stability (r=0.370, p<0.001), locomotor skills (r=0.271, p<0.001), and manipulative skills (r=0.165, p=0.002). In contrast, a slightly stronger correlation was observed for total MC (r=0.290, p<0.001), indicating partial alignment between teachers' estimates and students' actual performance.

Additionally, the Bland-Altman plot (Fig 3) assesses the consistency of the differences across performance. Results indicate a wide range of performance, suggesting considerable individual variability in teachers' estimations, with both underestimations and overestimations observed across all performance levels.

## Discussion

The purpose of this investigation was to assess the accuracy of PE teachers' assessments by comparing their PMC scores with students' AMC scores. Contrary to the initial hypothesis, PE teachers' perceptions were not always accurate, as evidenced by both under- and overestimations. Overall, moderate correlations were found between perceived and

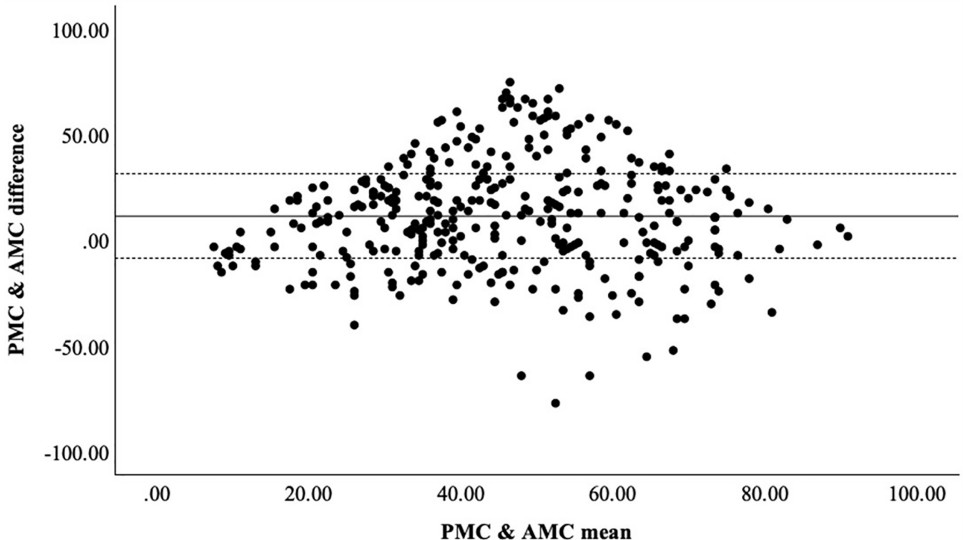

**Fig 3. Plot of differences between PMC and AMC vs. the mean of the two measurements.**

AMC, with the strongest link for stability skills, indicating that PE teachers have a general sense of students' abilities but do not consistently match those abilities with AMC.

The low correlation for the manipulative component likely reflects teachers' difficulty in estimating specific performance metrics, such as speed. Although manipulative skills are practiced in PE, assessments typically emphasize movement quality or task success rather than precise quantitative measures, leading to inconsistent evaluations. Bland-Altman analysis further highlights the wide variation between PMC and AMC. This highlights inconsistencies in teachers' judgments and underscores the need for structured, objective assessments in PE to ensure more accurate and fair evaluations.

Our results showed that teachers, in general, significantly overestimated almost all MC tests and components. Looking specifically at the main components, the most pronounced discrepancy was in stability, potentially due to the SP test, which differs considerably from typical PE exercises, making it harder for teachers to measure students' ACM. These results do not agree with the literature's general findings. For example, Ruiz et al. [34] found that PE teachers can accurately identify pupils' MC, even those with lower MC levels. In addition, Estevan et al. [35] also showed that the PMC of the PE teachers and the AMC of their students were correlated. Our findings may suggest that stability-related tasks (kinematic assessments), which frequently involve assessments of balance and coordination beyond the scope of typical dynamic activities observed in PE classes, require a more complex level of comprehension.

The overestimations might also reflect overconfidence in students' execution of these skills, especially in the locomotion and manipulation components, which are emphasized throughout the academic year and are central to the PE curriculum in Portugal [11]. Another possible explanation is that teachers rely heavily on contextual performance cues, such as students' behavior or participation in sports, rather than on objective evaluations of their motor skills. This confidence and their expectations [12,13] might bias their perceptions, especially when motor assessments are not routinely or formally conducted. Another important factor to consider is the long-standing reliance of PE teachers on assessing mastery levels – introductory, elementary, and advanced [36]. Such frameworks may have influenced their ability to adapt to the requirements of this investigation, which involved different observational criteria. Teachers may have found the tasks significantly different from those they were familiar with assessing, adding complexity to their evaluations.

Conversely, PE teachers significantly underestimated students' performance in the SLJ test, a component of the FitEscola® program, which is part of the PE evaluations in Portuguese schools [37]. It is important to emphasize that this was the only test in which the students scored above the average Portuguese normative values, which may have influenced their assessments. Also, such underestimation may be attributed to various factors, including the inherent subjectivity of individual perceptions, teachers' expectations, and preconceived notions about students' competence [12,38]. Future interventions could use visual aids or standardized benchmarks to calibrate teacher expectations. However, some studies have shown that teachers can identify children with low MC and are often the first to detect such cases. Nevertheless, few studies have analyzed this issue among PE teachers [39].

Our findings have key implications for PE teachers and school evaluations. Accurate assessment of students' MC is crucial for setting learning goals aligned with the PASEO framework [Profile at the end of compulsory education] [36]. Discrepancies between teachers' perceptions and AMC can lead to misaligned planning and unrealistic goals, affecting motivation and progress. Overestimation may lower students' self-confidence, while underestimation can reduce engagement and limit skill development. PE teachers should establish clear goals for observing and evaluating students' skills. Research shows that specialist PE teachers assess motor skills more effectively than generalist teachers, mainly due to their personal experience [40]. Therefore, teacher training programs should include motor skill assessment (kinematic assessments) components that provide tools and techniques for accurate, objective evaluations. Using standardized assessments can further reduce subjectivity and improve evaluation accuracy.

Despite the study's key findings, some limitations should be noted. Teachers had no prior experience with the test, and video demonstrations may have influenced their perceptions. Some tasks, especially the SHR (timed in seconds), were difficult to estimate. Another significant methodological consideration concerns the timing and design of the assessment.

The data collection period occurred in April 2024 and April 2025, corresponding to the final period of each academic year. Consequently, PE teachers had approximately eight months of uninterrupted contact with their students before providing their estimates. This design choice was intentional, as it reduced the likelihood that perceptual errors would result from teachers' unfamiliarity with students' motor abilities. Nonetheless, this decision is indicative of both a strength and a limitation. Despite the teachers' experience with their students at the time of assessment, it is not possible to determine whether the observed over- and underestimations reflect stable perceptual biases or temporary imprecisions. Finally, the study did not account for students' extracurricular sports participation, sex differences, or teachers' professional experience.

It seems imperative to acknowledge that discrepancies between perceived and AMC do not necessarily constitute detrimental outcomes, particularly in the short term. For instance, an initial underestimation of students' MC at the beginning of the academic year may enable PE teachers to establish realistic goals, provide adequate learning support, and incrementally enhance task difficulty. This approach can foster motivation and facilitate the acquisition of motor skills. From a pedagogical perspective, such cautious calibration may help teachers create learning environments that emphasize mastery, reduce early failure experiences, and promote perceptions of competence, especially among students with heterogeneous skill levels.

Little is known about PE teachers' PMC judgments, making this an important subject for future research that could explore, in our opinion, how biases are influenced by teachers' perceptions of students' socioeconomic status or other demographic factors, as these elements might also affect their estimations of competence [9]. Moreover, PE teachers' perceptions of other variables, such as physical fitness and technique across different teams and individual sports, should also be explored. Additionally, further research could investigate the longitudinal impact of misaligned perceptions on students' MC and engagement in physical activity. Also, future investigations might examine the effect of professional experience during these assessments. Finally, assessing teachers' PMC across educational levels, including classroom teachers, is also essential to enhancing the quality of schools' teaching, learning, and assessment processes.

## Practical implications

The present findings highlight the importance of providing targeted training for PE teachers, not only using FitEscola®, but also in assessing students' MC. Accurate evaluation is not only essential for aligning instruction with students' actual skill levels but also for fostering motivation and engagement by satisfying their basic psychological needs, as described by self-determination theory [1]. When PE teachers are equipped to identify students' MC, they can design learning environments (not only classes) that present optimally challenging tasks, promote individual responsibility, and encourage social connection through cooperative learning. Moreover, well-trained teachers who plan lessons based on valid assessments are better able to structure activities that correspond to what students are capable of achieving, thereby reducing frustration and disengagement. Such alignment between assessment, planning, and instructional practice ensures that PE classes remain inclusive, motivating, and developmentally appropriate. Therefore, teachers not only enhance students' MC but also strengthen their intrinsic motivation and sense of belonging, contributing to long-term participation in physical activity. Finally, investing in professional development programs that add motor assessment literacy and evidence-based instructional design is a key strategy for improving both the quality and equity of PE education.

## Conclusion

This research underscores the need for greater emphasis on motor assessment training in PE teacher education programs. It highlights the potential of integrating objective assessment methodologies to improve the accuracy of teachers' perceptions of their students' MC. These efforts are necessary to ensure that all students can fully develop their motor skills. Consequently, the misaligned perceptions of PE teachers should be viewed with concern, particularly given that data collection occurred mid-academic year, following the diagnostic assessment period.

## Supporting information

**S1 File. Inclusivity-in-global-research-questionnaire.**
(DOCX)

## Author contributions

**Conceptualization:** Fábio Flôres, Rita Cordovil.

**Data curation:** Fábio Flôres, Andre Pombo, Denise Soares, Dimitar Shabanliyski.

**Formal analysis:** Joana Serpa, Andre Pombo.

**Funding acquisition:** Denise Soares, Dimitar Shabanliyski.

**Investigation:** Joana Serpa, Rita Cordovil.

**Methodology:** Joana Serpa, Rita Cordovil.

**Resources:** Rita Cordovil.

**Validation:** Fernando Vieira.

**Writing – original draft:** Fábio Flôres, Joana Serpa, Fernando Vieira, Andre Pombo.

**Writing – review & editing:** Fernando Vieira, Andre Pombo, Denise Soares, Dimitar Shabanliyski.

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
