## [Decision Letter · Decision Letter 0]

13 Jan 2026

PONE-D-25-64838Are Teachers Missing the Beat on Students' Motor Competence?PLOS One?

Dear Dr. Soares,

Thank you for submitting your manuscript to PLOS ONE. After careful consideration, we feel that it has merit but does not fully meet PLOS ONE’s publication criteria as it currently stands. Therefore, we invite you to submit a revised version of the manuscript that addresses the points raised during the review process.

We look forward to receiving your revised manuscript.

Kind regards,

Gustavo De Conti Teixeira Costa, Ph.D

Academic Editor

PLOS One

**Journal Requirements:**

1. When submitting your revision, we need you to address these additional requirements. Please ensure that your manuscript meets PLOS ONE's style requirements, including those for file naming. The PLOS ONE style templates can be found at https://journals.plos.org/plosone/s/file?id=wjVg/PLOSOne_formatting_sample_main_body.pdf and https://journals.plos.org/plosone/s/file?id=ba62/PLOSOne_formatting_sample_title_authors_affiliations.pdf 2. Please include a complete copy of PLOS’ questionnaire on inclusivity in global research in your revised manuscript. Our policy for research in this area aims to improve transparency in the reporting of research performed outside of researchers’ own country or community. The policy applies to researchers who have travelled to a different country to conduct research, research with Indigenous populations or their lands, and research on cultural artefacts. The questionnaire can also be requested at the journal’s discretion for any other submissions, even if these conditions are not met.  Please find more information on the policy and a link to download a blank copy of the questionnaire here: https://journals.plos.org/plosone/s/best-practices-in-research-reporting. Please upload a completed version of your questionnaire as Supporting Information when you resubmit your manuscript. 3. Please provide additional details regarding participant consent. In the ethics statement in the Methods and online submission information, please ensure that you have specified (a) whether consent was informed and (b) what type you obtained (for instance, written or verbal, and if verbal, how it was documented and witnessed). If your study included minors, state whether you obtained consent from parents or guardians. If the need for consent was waived by the ethics committee, please include this information. If you are reporting a retrospective study of medical records or archived samples, please ensure that you have discussed whether all data were fully anonymized before you accessed them and/or whether the IRB or ethics committee waived the requirement for informed consent. If patients provided informed written consent to have data from their medical records used in research, please include this information. 4. In the online submission form, you indicated that your data will be submitted to a repository upon acceptance.  We strongly recommend all authors deposit their data before acceptance, as the process can be lengthy and hold up publication timelines. Please note that, though access restrictions are acceptable now, your entire minimal  dataset will need to be made freely accessible if your manuscript is accepted for publication. This policy applies to all data except where public deposition would breach compliance with the protocol approved by your research ethics board. If you are unable to adhere to our open data policy, please kindly revise your statement to explain your reasoning and we will seek the editor's input on an exemption. 5. When completing the data availability statement of the submission form, you indicated that you will make your data available on acceptance. We strongly recommend all authors decide on a data sharing plan before acceptance, as the process can be lengthy and hold up publication timelines. Please note that, though access restrictions are acceptable now, your entire data will need to be made freely accessible if your manuscript is accepted for publication. This policy applies to all data except where public deposition would breach compliance with the protocol approved by your research ethics board. If you are unable to adhere to our open data policy, please kindly revise your statement to explain your reasoning and we will seek the editor's input on an exemption. Please be assured that, once you have provided your new statement, the assessment of your exemption will not hold up the peer review process. 6. Please upload a new copy of Figures 2 and 3, as the detail is not clear. Please follow the link for more information:  https://journals.plos.org/plosone/s/figures 7. Please include your tables as part of your main manuscript and remove the individual files. Please note that supplementary tables (should remain/ be uploaded) as separate "supporting information" files. 8. If the reviewer comments include a recommendation to cite specific previously published works, please review and evaluate these publications to determine whether they are relevant and should be cited. There is no requirement to cite these works unless the editor has indicated otherwise. 

**Additional Editor Comments:**

Dear authors, good morning! Thank you for submitting the manuscript to PLOS One. After peer review, please make the necessary changes and send a point-by-point response letter. Thank you for submitting it, and I await your reply.

Sincerely,

Gustavo De Conti Teixeira Costa

Reviewers' comments:

Reviewer's Responses to Questions

**Comments to the Author**

1. Is the manuscript technically sound, and do the data support the conclusions?

Reviewer #1: Yes

Reviewer #2: Yes

Reviewer #3: Partly

2. Has the statistical analysis been performed appropriately and rigorously?

Reviewer #1: Yes

Reviewer #2: Yes

Reviewer #3: Yes

3. Have the authors made all data underlying the findings in their manuscript fully available?

Reviewer #1: Yes

Reviewer #2: Yes

Reviewer #3: Yes

4. Is the manuscript presented in an intelligible fashion and written in standard English?

Reviewer #1: Yes

Reviewer #2: Yes

Reviewer #3: Yes

**Reviewer #1:** Dear authors, please pay attention to the items mentioned in the different sections of the article, such as Abstract, Introduction, Method, Discussion, and try to review the items mentioned and apply the points of view comprehensively and completely. I hope that by completing these sections, your article will be more complete.

**Reviewer #2:** Dear authors,

I find this work to contain excellent rigor and the methodology is sound. I do however has a list of suggestion to help improve the quality of the work, set up future directions, and explain the limitations more thoroughly. Below are my suggestions:

• Abstract and purpose statement in the paper: The purpose statement states to compare PE teachers perceptions of their students’ motor competence. However, it does not state what it is comparing this to. Although you explain that you are comparing PMC to the actual motor competence in the methods, your purpose statement in the abstract and introduction does not grammatically state this. I suggest a revisions to make this statement stronger. For example: “The purpose was to compare physical education teachers perception of their students motor competence with the students actual motor competence”. Please do this for both the abstract and introduction.

• Introduction line 81: There are typos in the sentence “Accurate diagnostic. Formative. And summative assessment” I believe you meant to have commas not periods.

• Methods-Sample: How long have the teachers worked with their group of students? Was this the start of their work with them, have they been working with the students for multiple semester/years? Is this data unknown? Adding this decretive information would be informative to understanding if the data is showing PE teachers perceptions at the start of their work with the students, during their work with the students, or after several years with the students. I will have more comments on this point for other sections.

• Methods-Data analysis: The Bland-Altman plots are a very strong inclusion for the paper. However, I disagree with the execution of having all the motor subsets in one figure/ analysis. Bland-alman plots allow you to see where under/over estimation are occurring, and if the assessment is uniform across the whole scale of the assessment tool. For example, in the current figure (figure 3), we can see that at the lower and upper limits, PE are great at assessing low and above average performance. However, the middle of the assessment is poor. I suggest, separating the multiple subsets and making the analysis more complete and informative for each subset.

• Figures 2 and 3. Both of these figures need more descriptive information. In figure 2, the readers do not know what the error bars are. Are these standard deviation, standard error, or something else? In figure 3 we do not know what the dashed and solid lines are. In addition, since multiple subsets of data are in the plot, the reader do not know if under/over estimation are occurring consistently for a single participant (i.e. individual participants are clusters in the plot) or if different subsets have different patterns in the tool. I suggest multiple plots for the different skills being presented.

• Discussion: One limitation in the discussion is that we do not know how long the PE teachers have been working with the students and that these data are a single time point. If the assessment occurred at the start of the PE teachers work with the students, then error is more likely to occur since they are a novice with this group of students. Additionally, if the PE teacher has been working with the students for a long time, then under and over estimation of skills might be more detrimental, because biases might have been occurring for a long time. Last, since this is a single time point, we do not know if teachers are consistently over and under estimating student performance, a longitudinal assessment would be an excellent next step to see if teachers are consistently perceiving skills inaccurately, or if they get better with time.

• Discussion: While I agree with the point that over and underestimating skills can be detrimental to Physical education, I think the nuance that consistent over and underestimation needs to be explained further. For example, underestimation at the start of PE class might be beneficial, since PE teachers can set up realistic goals that continue to challenge students. I think these nuances should be discussed and included in the discussion.

• Discussion: When comparing your work to others, have other studies factored in the time PE teachers work with students and how this affects their perception of MC? If it has, this should be included. If it hasn’t then this is an excellent point to include for future work when you perform a longitudinal study.

**Reviewer #3:** I am glad to have had the opportunity to review this manuscript, as it addresses an important and up-to-date issue.

Understanding PE teachers’ perceptions of students’ motor competence is undoubtedly valuable, as these perceptions directly guide daily teaching–learning processes. However, I have some concerns regarding the interpretation of the findings, which I present in the attached document.

**Do you want your identity to be public for this peer review?** For information about this choice, including consent withdrawal, please see our Privacy Policy

Reviewer #1: No

Reviewer #2: **Yes:** Marcelo Rosales

Reviewer #3: **Yes:** Claudio M. F. Leite

---

## [Author Response · Author response to Decision Letter 1]

21 Jan 2026

Dear Editor,

We appreciate the opportunity to improve the quality of our work. We followed the commentaries made by the reviewers. Answers were provided in red in this file and the manuscript for better clarity.

Reviewer #1: Dear authors, please pay attention to the items mentioned in the different sections of the article, such as Abstract, Introduction, Method, Discussion, and try to review the items mentioned and apply the points of view comprehensively and completely. I hope that by completing these sections, your article will be more complete.

Abstract: A brief reference to statistical methods is also provided.

R: Thank you for the suggestion. Improvements were made.

Introduction: The introduction should refer more to the research background, more to the theoretical foundations, and also to the importance of the work, which should be after the necessity of the research. I did not see any explanation in this regard.

R: We have revised the introduction to strengthen the theoretical background.

Methods: The entry criterion refers to the student's health, which was done by the instructor. Please indicate on what basis did the instructor assess the student's health? The validity and reliability of the tools are not mentioned.

R: We respectfully clarify that students’ health status was not clinically assessed by the instructor (or PE teacher) but verified through parental information routinely used in PE classes.

Discussion: This section needs revision. You should first summarize the results of your work and then logically review your findings with the studies you mentioned in the introduction section. Examine the results of your work in more depth with previous studies. Also, better articulate the research that supports and contradicts your results.

R: Thank you for the suggestion. We have revised the discussion section to improve its structure and depth.

Reviewer #2:

• Abstract and purpose statement in the paper: The purpose statement states to compare PE teachers perceptions of their students’ motor competence. However, it does not state what it is comparing this to. Although you explain that you are comparing PMC to the actual motor competence in the methods, your purpose statement in the abstract and introduction does not grammatically state this. I suggest a revisions to make this statement stronger. For example: “The purpose was to compare physical education teachers perception of their students motor competence with the students actual motor competence”. Please do this for both the abstract and introduction.

R: We agree with the reviewer and have revised the purpose statements in both the abstract and the introduction to explicitly state that teachers’ perceived motor competence was compared with students’ actual motor competence.

• Introduction line 81: There are typos in the sentence “Accurate diagnostic. Formative. And summative assessment” I believe you meant to have commas not periods.

R: Thank you for noticing the mistake. The sentence was corrected.

• Methods-Sample: How long have the teachers worked with their group of students? Was this the start of their work with them, have they been working with the students for multiple semester/years? Is this data unknown? Adding this decretive information would be informative to understanding if the data is showing PE teachers perceptions at the start of their work with the students, during their work with the students, or after several years with the students. I will have more comments on this point for other sections.

R: We have added descriptive information to the methods section specifying the duration of teacher-student contact prior to data collection.

• Methods-Data analysis: The Bland-Altman plots are a very strong inclusion for the paper. However, I disagree with the execution of having all the motor subsets in one figure/ analysis. Bland-alman plots allow you to see where under/over estimation are occurring, and if the assessment is uniform across the whole scale of the assessment tool. For example, in the current figure (figure 3), we can see that at the lower and upper limits, PE are great at assessing low and above average performance. However, the middle of the assessment is poor. I suggest, separating the multiple subsets and making the analysis more complete and informative for each subset.

R: The present analysis was intentionally conducted using a single Bland-Altman plot to provide an overall representation of agreement between perceived and AMC, consistent with the study’s goal. We fully agree that subset-specific Bland-Altman analyses represent an excellent direction for future research and could offer further insight into domain-specific perceptual biases.

• Figures 2 and 3. Both of these figures need more descriptive information. In figure 2, the readers do not know what the error bars are. Are these standard deviation, standard error, or something else? In figure 3 we do not know what the dashed and solid lines are. In addition, since multiple subsets of data are in the plot, the reader do not know if under/over estimation are occurring consistently for a single participant (i.e. individual participants are clusters in the plot) or if different subsets have different patterns in the tool. I suggest multiple plots for the different skills being presented.

R: We have revised the figure captions to provide clearer and more explicit descriptive information regarding the error bars in Figure 2 and the reference lines in Figure 3, thereby improving interpretability. Regarding the suggestion to present multiple Bland-Altman plots by motor skill subsets, we agree that this approach could provide additional insight into domain-specific estimation patterns and consider it an important direction for future research; however, the current figure was intentionally designed to illustrate overall agreement patterns rather than individual- or subset-specific clustering.

• Discussion: One limitation in the discussion is that we do not know how long the PE teachers have been working with the students and that these data are a single time point. If the assessment occurred at the start of the PE teachers work with the students, then error is more likely to occur since they are a novice with this group of students. Additionally, if the PE teacher has been working with the students for a long time, then under and over estimation of skills might be more detrimental, because biases might have been occurring for a long time. Last, since this is a single time point, we do not know if teachers are consistently over and under estimating student performance, a longitudinal assessment would be an excellent next step to see if teachers are consistently perceiving skills inaccurately, or if they get better with time.

R: We appreciate this comment and clarify that data collection was intentionally conducted near the end of the academic year to ensure that teachers had prolonged exposure to their students (April, as mentioned in the methods section). Nevertheless, we acknowledge that the cross-sectional design limits conclusions regarding the stability of teachers’ perceptual biases over time, which we now explicitly address as a limitation and future research direction.

• Discussion: While I agree with the point that over and underestimating skills can be detrimental to Physical education, I think the nuance that consistent over and underestimation needs to be explained further. For example, underestimation at the start of PE class might be beneficial, since PE teachers can set up realistic goals that continue to challenge students. I think these nuances should be discussed and included in the discussion.

R: We agree with the reviewer and have expanded the Discussion to better articulate the nuanced effects of over- and underestimation of motor competence.

• Discussion: When comparing your work to others, have other studies factored in the time PE teachers work with students and how this affects their perception of MC? If it has, this should be included. If it hasn’t then this is an excellent point to include for future work when you perform a longitudinal study.

R: To the best of our knowledge, no previous studies have examined PE teachers’ PMC using a direct comparison with objectively assessed MC, as stated in the Introduction. Consequently, existing research has not accounted for the duration of teacher-student interaction as a factor influencing perceptual accuracy. We agree that investigating how teachers’ perceptions evolve over time represents an important and novel direction for future longitudinal research.

Reviewer #3:

Major Issues

Previous studies cited by the authors (e.g., Ruiz et al., 2001; Estevan et al., 2018) indicate that PE teachers can perceive students’ motor competence with certain degree of accuracy when it is referenced to movement patterns or qualitative process measures. In contrast, the present study relies on kinematic variables (e.g., time, velocity), which are less familiar in to PE teachers and in pedagogical contexts in general, and may require specific training for meaningful interpretation.

R: We agree with the reviewer’s observation, and this distinction is already addressed in the Discussion and Limitations sections of the manuscript. We note that previous studies focused on qualitative movement patterns, whereas the present study used objective, quantitative performance measures (e.g., time and velocity), which may be less familiar to PE teachers and require specific training for accurate interpretation.

Thus, the observed discrepancy between teachers’ perceptions and test outcomes may not reflect a limitation in teachers’ ability to assess motor competence per se, but rather a difficulty in interpreting novel kinematic-based indicators. This distinction is central to the manuscript and should have been clearly raised from the introduction but is only briefly acknowledged in the limitations.

Recent evidence suggests that, although kinematic variables may be strongly associated with performance outcomes, different kinematic indicators can lead to distinct interpretations of motor competence and would benefit from being examined alongside movement pattern analyses (e.g., Leite et al., 2025), which is the clearer reference we have for MC. This perspective may help contextualize the present findings and avoid overgeneralization regarding assessment accuracy in physical education.

R: We agree with the reviewer’s interpretation. Thank you for the suggestion; the changes have been made in the introduction and discussion sections.

Table 1 clarifies that teachers were asked to estimate performance outcomes expressed in specific quantitative variables (e.g., Kinematics, units, products), rather than to judge movement quality or execution patterns. This procedural detail strengthens the study’s transparency but also highlights a key interpretative issue: the observed discrepancies may reflect unfamiliarity with the metrics and units employed, rather than limited ability to perceive students’ motor competence per se; even with an ‘average child reference’.

R: Thank you for the suggestion. Changes have been made accordingly.

Therefore, I recommend that the authors:

1. Adopt more cautious language throughout the manuscript, (from the abstract). Statements suggesting a general lack of assessment accuracy or the need for curricular adjustment should be attenuated and explicitly tied to the specific type of test and variables used, including “the level” of familiarity of the PE teachers with them.

R: Thank you for the suggestion, changes were made throughout the manuscript.

2. Clarify the pedagogical meaning of the kinematic variables employed, emphasizing that accurate use of such measures likely requires targeted familiarization or training. This would strengthen the argument for professional development without implying deficiencies in teachers’ general assessment competence.

R: Thank you for the suggestion, changes were made.

3. Further articulate the value of the selected tests by explaining how kinematic-based measures complement or are articulated to more traditional process-oriented assessments.

Overall, with clearer conceptual framing and moderated claims, this study has the potential to make a meaningful contribution to the literature on motor competence and its assessment in physical education.

R: Thank you for the suggestion, changes were made.

Minor issue:

1. Check for typos and punctuations throughout the text.

R: Done.

2. Lines 95-100 conclude the introduction and point to the aim but do not specify it clearly as in the abstract. Check this when rewriting.

R: Done.

3. Line 104 says three schools and line 112 says 5. Check this.

R: Done.

4. Line 105 reads (2023–2024). I supose it is years. Specify it.

R: Done.

5. Lines 113. Specify how inclusion criteria were assessed.

R: Done.

6. Line 175. “outliers were checked and corrected or removed as needed.” Specify the criteria.

R: Done.

7. Lines 177-179. “Cohen's d ((mean 1 – mean 2)/pooled standard deviation) as the index of effect size (large - η2 > 0.8; moderate - η2 between 0.8 and 0.5; small η2 between 0.49 and 0.20; trivial - η2 < 0.2) (Cohen, 2013).” Consider rephrasing it.

R: Done.

8. Lines 181-183: “The estimation was considered accurate when the difference was lower than 5%. Differences of more than 5% were considered overestimations if the estimate exceeded performance”. Explain better the reason of this 5%.

R: Thank you for this comment and for the opportunity to clarify our criterion. In our study, a 5% difference between the teacher’s estimation and the child’s AMC was considered an acceptable margin of error rather than requiring a perfectly exact match (0% error). This decision was based on two main reasons: (i) in practical school settings it is extremely difficult for teachers to estimate motor performance with absolute precision in specific MCA tasks, and (ii) previous studies on perceived and AMC have also adopted small error ranges to classify estimates as accurate or inaccurate (e.g., Almeida et al., 2017). This tolerance allows us to distinguish meaningful overestimations from minor, non‑substantial deviations that are inevitable when teachers judge performance in real-world conditions. We have now clarified this rationale in the Methods section.

Almeida, G., Luz, C., Martins, R., & Cordovil, R. (2017). Do children accurately estimate their performance of fundamental movement skills? Journal of Motor Learning and Development, 5(2), 193–206. https://doi.org/10.1123/jmld.2016-0030

9. Line 191. Gpower has a specific calculation for SPSS packeage. Was it considered? If so, state it in the sample session.

R: The G*Power calculation was conducted using the statistical test corresponding to the paired-samples t-test, which is consistent with the analyses performed in SPSS and was described in the sample section.

10. Line 214. I thinks it is “most teachers” instead of “more teacher”.

R: Done.

11. Figures. Enhance the quality of figure 1, and figure 2 is cut in the end (MC).

R: Done.

---

## [Decision Letter · Decision Letter 1]

14 Feb 2026

Are Teachers Missing the Beat on Students' Motor Competence?PLOS One?

Dear Dr. Soares,

Thank you for submitting your manuscript to PLOS ONE. After careful consideration, we feel that it has merit but does not fully meet PLOS ONE’s publication criteria as it currently stands. Therefore, we invite you to submit a revised version of the manuscript that addresses the points raised during the review process.

We look forward to receiving your revised manuscript.

Kind regards,

Gustavo De Conti Teixeira Costa, Ph.D

Academic Editor

PLOS One

Journal Requirements:

Additional Editor Comments:

Dear authors, thank you for submitting the revised manuscript. To ensure we reach the end of this editorial process, please pay attention to the reviewer's suggestions for minor corrections. I believe these adjustments will improve the quality of the final version of the work.

Reviewers' comments:

Reviewer's Responses to Questions

**Comments to the Author**

Reviewer #1: All comments have been addressed

Reviewer #2: All comments have been addressed

Reviewer #3: All comments have been addressed

2. Is the manuscript technically sound, and do the data support the conclusions?

Reviewer #1: Yes

Reviewer #2: Yes

Reviewer #3: Partly

3. Has the statistical analysis been performed appropriately and rigorously?

Reviewer #1: Yes

Reviewer #2: Yes

Reviewer #3: Yes

4. Have the authors made all data underlying the findings in their manuscript fully available?

Reviewer #1: Yes

Reviewer #2: Yes

Reviewer #3: Yes

5. Is the manuscript presented in an intelligible fashion and written in standard English?

Reviewer #1: Yes

Reviewer #2: Yes

Reviewer #3: Yes

Reviewer #1: Thank you to the respected authors who paid due attention to the comments made by the referee, completed them, and implemented each one correctly.

Reviewer #2: Dear authors,

Thank you for addressing all of my comments! All of my comments have been addressed.

Great work!

Reviewer #3: General comment

I would like to thank the authors for their careful and thoughtful revision of the manuscript. The changes made throughout the text have substantially improved the conceptual framing and clarity of the study.

In particular, the interpretation of the findings is now more cautious and appropriately contextualized. The manuscript no longer suggests a generalized limitation in teachers’ assessment accuracy but instead acknowledges the possible interpretative challenges associated with performance-based and kinematic metrics. This clarification addresses the primary concern raised in the previous review and strengthens the overall contribution of the study. However, I still have some minor issues; some are new, and some are not completely solved issues.

Minor issues

1. Paragraph structure (Lines 103–120)

Lines 103-120 appear to contain more than one conceptual block and may benefit from being divided into two or three paragraphs. Revising this structure would improve clarity and readability. Check it out.

2. Conceptual clarity (Lines 112–113)

The sentence “Therefore, to the best of our knowledge, no other research has examined the accuracy of kinematic measurements based on PE teachers' perceptions of their students' MC.” appears conceptually imprecise.

Kinematic measurements are objective variables; what is being examined is the accuracy of teachers’ estimations relative to those objective measurements. I recommend revising the sentence to avoid potential misunderstanding.

Suggested revision: (Just a suggestion. See if it makes sense)

“To the best of our knowledge, no previous research has examined the accuracy of PE teachers’ estimations of students’ motor competence when compared to objective kinematic measurements.”

3. Outlier handling not solved (Line 212)

The manuscript states that extreme statistical outliers were checked and corrected or removed as needed (Rodrigues et al., 2025). While removing outliers can be appropriate, the criteria used to identify and manage them are not clearly specified.

I recommend explicitly stating the criteria adopted (e.g., statistical thresholds, influence diagnostics, standard deviation cutoffs) and clarifying whether removal or correction affected the results. This would enhance transparency and reproducibility.

4. Justification of 5% tolerance not solved (Lines 219–222)

The manuscript indicates that a tolerance of 5% was adopted (Almeida et al., 2017). However, Almeida et al. (2017) does not appear to provide a clear justification for this specific threshold.

Please clarify the rationale for selecting 5% rather than another threshold (e.g. 3%, 7%, 10%) Was it a theoretical basis, empirical precedent, or practical reasoning? If the choice was based on researcher judgment, this should be explicitly acknowledged to allow critical appraisal.

5. Figure quality not solved (Figures 1 and 2)

Figures 1 and 2 remain low in resolution and may not meet publication-quality standards. Please provide higher-resolution versions. Additionally, check for typos (e.g., “Locomtor” instead of “Locomotor” in Figure 1).

6. Minor typos

Please review the manuscript thoroughly for minor typos (e.g., “inApril 2024”).

**Do you want your identity to be public for this peer review?** For information about this choice, including consent withdrawal, please see our Privacy Policy

Reviewer #1: No

Reviewer #2: **Yes:** Marcelo Rosales

Reviewer #3: **Yes:** Claudio Manoel Ferreira Leite

---

## [Author Response · Author response to Decision Letter 2]

16 Feb 2026

Dear Editor,

We appreciate the new opportunity to improve the quality of our work. We followed the reviewers’ comments. Answers were provided in red in this file and the manuscript for better clarity.

Reviewer #3: General comment

1. Paragraph structure (Lines 103–120)

Lines 103-120 appear to contain more than one conceptual block and may benefit from being divided into two or three paragraphs. Revising this structure would improve clarity and readability. Check it out.

R: Done.

2. Conceptual clarity (Lines 112–113)

The sentence “Therefore, to the best of our knowledge, no other research has examined the accuracy of kinematic measurements based on PE teachers’ perceptions of their students’ MC.” appears conceptually imprecise.

Kinematic measurements are objective variables; what is being examined is the accuracy of teachers’ estimations relative to those objective measurements. I recommend revising the sentence to avoid potential misunderstanding.

Suggested revision: (Just a suggestion. See if it makes sense)

“To the best of our knowledge, no previous research has examined the accuracy of PE teachers’ estimations of students’ motor competence when compared to objective kinematic measurements.”

R: Thank you for the suggestion. Changes were performed accordingly.

3. Outlier handling not solved (Line 212)

The manuscript states that extreme statistical outliers were checked and corrected or removed as needed (Rodrigues et al., 2025). While removing outliers can be appropriate, the criteria used to identify and manage them are not clearly specified.

I recommend explicitly stating the criteria adopted (e.g., statistical thresholds, influence diagnostics, standard deviation cutoffs) and clarifying whether removal or correction affected the results. This would enhance transparency and reproducibility.

R: Thank you for the suggestion. Changes were performed accordingly, and we also added another investigation that supports our choice.

4. Justification of 5% tolerance not solved (Lines 219–222)

The manuscript indicates that a tolerance of 5% was adopted (Almeida et al., 2017). However, Almeida et al. (2017) does not appear to provide a clear justification for this specific threshold.

Please clarify the rationale for selecting 5% rather than another threshold (e.g. 3%, 7%, 10%) Was it a theoretical basis, empirical precedent, or practical reasoning? If the choice was based on researcher judgment, this should be explicitly acknowledged to allow critical appraisal.

R: We appreciate the feedback. We agree that Almeida et al. (2017) do not provide a formal theoretical explanation for using a 5% tolerance threshold. In our study, we chose the 5% criterion as a practical way to define meaningful differences in how people perceive things. This was done while also considering how difficult it is to estimate quantitative motor performance outcomes (e.g., time or velocity) in PE settings. We have updated the manuscript to more clearly state that this limit was based on practical considerations and is consistent with previous methods used in research on perceived versus actual competence. However, we also acknowledge that different limits could lead to slightly different classifications.

5. Figure quality not solved (Figures 1 and 2)

Figures 1 and 2 remain low in resolution and may not meet publication-quality standards. Please provide higher-resolution versions. Additionally, check for typos (e.g., “Locomtor” instead of “Locomotor” in Figure 1).

R: Thank you. The figures in the manuscript are high resolution, and we believe the reduced quality may be due to the PDF conversion or the viewing format. Additionally, we carefully reviewed the entire manuscript to identify and correct any typos or other writing inaccuracies. Thank you for bringing this to our attention.

6. Minor typos

Please review the manuscript thoroughly for minor typos (e.g., “inApril 2024”).

R: Done.

---

## [Editor Report · Decision Letter 2]

18 Feb 2026

Are Teachers Missing the Beat on Students' Motor Competence?

PONE-D-25-64838R2

Dear Dr. Soares,

We’re pleased to inform you that your manuscript has been judged scientifically suitable for publication and will be formally accepted for publication once it meets all outstanding technical requirements.

Kind regards,

Gustavo De Conti Teixeira Costa, Ph.D

Academic Editor

PLOS One

Additional Editor Comments (optional):

Dear authors, I hope this message finds you well. I am pleased to congratulate you on your work. It is with great satisfaction that I inform you that the manuscript has been accepted for publication.

Sincerely, Gustavo.
---

## [Editor Report · Acceptance letter]

PONE-D-25-64838R2

PLOS One

Dear Dr. Soares,

I'm pleased to inform you that your manuscript has been deemed suitable for publication in PLOS One. Congratulations! Your manuscript is now being handed over to our production team.

Kind regards,

on behalf of

Dr. Gustavo De Conti Teixeira Costa

Academic Editor

PLOS One